# Antibodies Against Anti-Oxidant Enzymes in Autoimmune Glomerulonephritis and in Antibody-Mediated Graft Rejection

**DOI:** 10.3390/antiox13121519

**Published:** 2024-12-12

**Authors:** Maurizio Bruschi, Giovanni Candiano, Andrea Petretto, Andrea Angeletti, Pier Luigi Meroni, Marco Prunotto, Gian Marco Ghiggeri

**Affiliations:** 1Unit of Nephrology, Dialysis and Transplantation and Laboratory of Molecular Nephrology, Core Facilities-Proteomics Laboratory, 16147 Genoa, Italygiovannicandiano@gaslini.org (G.C.); andreaangeletti@gaslini.org (A.A.); 2Department of Experimental Medicine (DIMES), University of Genoa, 16132 Genoa, Italy; 3IRCCS Istituto Giannina Gaslini, 16147 Genoa, Italy; andreapetretto@gaslini.org; 4Experimental Laboratory of Immunological and Rheumatologic Researches, Istituto Auxologico Italiano–Istituto di Ricovero e Cura a Carattere Scientifico, 20149 Milano, Italy; pierluigi.meroni@unimi.it; 5Institute of Pharmaceutical Sciences of Western Switzerland, University of Geneva, 1205 Geneva, Switzerland

**Keywords:** oxidants, superoxide dismutase, glutathione, glutathione synthetase, autoimmune diseases, membranous nephropathy, antibody-mediated reaction

## Abstract

Historically, oxidants have been considered mechanisms of glomerulonephritis, but a direct cause–effect correlation has never been demonstrated. Several findings in the experimental model of autoimmune conditions with renal manifestations point to the up-regulation of an oxidant/anti-oxidant system after the initial deposition of autoantibodies in glomeruli. Traces of oxidants in glomeruli cannot be directly measured for their rapid metabolism, while indirect proof of their implications is derived from the observation that Superoxide Oxidase 2 (SOD2) is generated by podocytes after autoimmune stress. The up-regulation of other anti-oxidant systems takes place as well. Here, we discuss the concept that a second wave of antibodies targeting SOD2 is generated in autoimmune glomerulonephritis and may negatively influence the clinical outcome. Circulating and renal deposits of anti-SOD2 antibodies have been detected in patients with membranous nephropathy and lupus nephritis, two main examples of autoimmune disease of the kidney, which correlate with the clinical outcome. The presence of anti-SOD2 antibodies in circulation and in the kidney has been interpreted as a mechanism which modifies the normal tissue response to oxidative stress. Overall, these findings repropose the role of the oxidant/anti-oxidant balance in autoimmune glomerulonephritis. The same conclusion on the oxidant/anti-oxidant balance may be proposed in renal transplant. Patients receiving a renal graft may develop antibodies specific for Glutathione Synthetase (GST), which modulates the amount of GST disposable for rapid scavenging of reactive oxygen species (ROS). The presence of anti-GST antibodies in serum is a major cause of rejection. The perspective is to utilize molecules with known anti-oxidant effects to modulate the anti-oxidative response in autoimmune pathology of the kidney. A lot of molecules with known anti-oxidant effects can be utilized, many of which have already been proven effective in animal models of autoimmune glomerulonephritis. Many molecules with anti-oxidant activity are natural products; in some cases, they are constituents of diets. Owing to the simplicity of these drugs and the absence of important adverse effects, many anti-oxidants could be directly utilized in human beings.

## 1. Introduction

Oxidant generation is a prerequisite of living cells, representing an end by-product of cellular metabolic processes. Oxidants are produced following two different pathways: one is the respiratory chain, where electrons derived from NADH or FADH react with either oxygen or other molecules to generate unstable, reactive oxygen species (ROS); the second origin is enzymatic from NADH oxidases, a multi-enzymatic complex with variable distributions in tissues, which catalyze the transport of one electron from NADPH to oxygen. Both systems generate unstable molecules, such as O_2_^−^, H_2_O_2_, and OH^−^, that freely react with proteins and lipids, including constituents of biological membranes [1].

Studies of cardiovascular diseases and cancer have greatly contributed to extending the basic concepts about the importance of redox homeostasis in human pathologies [2,3]. They pointed to the crucial role of a stable balance between ROS generation and scavenging [3]. The focus is the multi-system of detoxifying molecules containing thiol groups, such as glutathione (GSH) and enzymes transforming reactive molecules in neutral compounds, that together constitute the basis of scavenging. Superoxide dismutase, which converts superoxide O_2_^−^ to H_2_O_2_, and Glutathione Synthetase (GST), which modulates the amount of GSH which can be disposed of for rapid scavenging of ROS, are attracting major interest. All of these molecules have been historically studied in renal pathologies without obtaining significant results.

In this perspective, we propose that in autoimmune pathologies of the kidney, the interest in ROS has been renewed based on the observation that ROS represent a second trigger of autoimmunity and have adjunctive effects on the progression of renal damage. There is evidence that in autoimmune glomerulonephritis, in synergy with the deposition of autoantibodies in glomeruli (the first hit of the disease), the synthesis of SOD2 is up-regulated in podocytes, which is an indirect sign of oxidative stress. In some patients, and in parallel with SOD2 up-regulation, anti-SOD2 antibodies are generated and can be detected in the circulation and within glomeruli, where it potentially modifies the equilibrium of the anti-oxidant system. Therefore, anti-SOD2 antibodies may have a role as second-wave players in renal autoimmune conditions, in which other autoantibodies represent the start. A second important area of investigation is renal transplantation since recipients of renal grafts who develop Antibody-Mediated Rejection (AMR) have high circulating levels of anti-GST antibodies that correlate with the entity of the renal damage. In this case, anti-GST generation potentially reflects an increase in GST in kidneys, which represents an indirect signal of oxidative stress.

## 2. ROS and Renal Pathologies: A Historical View

ROS have been, for some time, considered a major driver of renal pathologies, especially in those conditions involving the slit-diaphragm, such as minimal change nephropathy (MCD) and Focal Segmental Glomerulosclerosis (FSGS). The proposed idea was that ROS modified the selectivity of the glomerular basement membrane versus circulating proteins and caused proteinuria; peroxidation by ROS of the lipid constituents of the membrane was considered the mechanism. Two reasons have supported this concept. The first piece of support derives from experimental models of proteinuric glomerular diseases induced by puromycin and adriamycin, which are molecules with oxidant activity [4,5]. The oxidant activity is particularly strong for adriamycin, an antitumoral drug that stimulates glomerular ROS synthesis via the xanthine pathway; in glomeruli deprived of xanthine oxidase, ROS generation after adriamycin exposure is practically nil. The implication of oxidants in both models of proteinuric renal pathologies has been investigated until recently, considering the potential beneficial effects of anti-oxidants. The second reason supporting a direct link between oxidants and proteinuria is the observation that knockout mice for the two major detoxifying enzymes, SOD and GST, develop spontaneous proteinuria and focal glomerular damage [6,7]. *Gstm1* knockout mice, in particular, display increased oxidative stress, kidney injury, and inflammation that can be rescued by treatment with drugs with dismutase function, such as Tempol [7].

In the absence of any significant evolutions, the interest in oxidants as factors involved in proteinuria has slowly declined. The discovery of the genetic basis of MCD/FSGS focused the interest of research on podocytes as the main cells governing the permeability properties of glomeruli; the result was a redefinition of the mechanisms of renal pathologies [8,9]. It has been shown, in particular, that a conspicuous quota of proteinuric diseases, and specifically MCD and FSGS, is caused by inherited genetic modification [10,11] in one of the many genes that govern podocyte protein trafficking.

## 3. ROS in Primary and Secondary Autoimmune Glomerulonephritis

### 3.1. General Concepts

Glomeruli are the renal site where immunologic and autoimmune events take place and may modify the structure of the basement membrane. Membranous nephropathy (MN) and Lupus nephritis (LN) are the two major autoimmune glomerulonephritis conditions causing proteinuria with frequent evolution to renal failure. Innate and adaptative immunity are drivers of the autoimmune process and ROS may be a part of the mechanism. Oxidants are produced by neutrophils and natural killers that constitute the barrier against invading parasites, bacteria, viruses, and others. The high metabolic request of resident cells following an autoimmune reaction represents, instead, the link between ROS and adaptative immunity. In this case, regulatory T cells that have a key part in adaptative immunity are predominantly supported by fatty oxidation and oxidative phosphorylation that generate ROS [12].

### 3.2. Membranous Nephropathy

MN is caused by the deposition of specific antibodies along the basement glomerular membrane (GBM) of glomeruli. Several proteins of the GBM have been discovered as target antigens in patients with MN; phospholipase A2-receptor (PLA2R1) [13] is the main autoantigen recognized in almost 60% of MN patients and Thrombospondin-type1-domain-7A (THSD7A) is the second antigen in frequency (2–5%) [14]. Other auto-antigens have been recently recognized as part of the autoimmune mechanism with a frequency that is, in almost all cases, <1% [15,16,17,18,19,20,21,22,23,24].

Indirect evidence supports the concept that ROS are activated in glomeruli following the deposition of autoantibodies. Buelli et al. [25] demonstrated that IgG4 purified from patients with MN induced either the synthesis or externalization of SOD by podocytes ‘in vitro’, which followed strong intracellular acidification. This phenomenon was interpreted as an indirect sign of increased oxidation. An extensive analysis of SOD expression and glomerular localization in bioptic renal tissues in MN patients demonstrated the granular peripheral SOD2 staining along the glomerular capillary wall in many of them; by comparison, SOD2 was not detectable in glomeruli of normal kidneys [25]. In line with this concept, Tomas et al. [26] showed that after 70 days from the injection of anti-THSD7A antibody-containing serum in mice, podocyte SOD2 expression was increased.

### 3.3. Lupus Nephritis

LN is determined by autoantibodies targeting several antigens in glomeruli; the main ones are dsDNA/anti-Enolase (ENO)/anti-Annexin A1 (ANXA1)/Histone A [27,28]. This is probably an incomplete list since, with the introduction of technologies based on peptide arrays, many more circulating antibodies have been characterized in LN patients and some of them target proteins of glomeruli [29]. Recent studies demonstrated the increase in markers of oxidative stress in glomeruli and urine in LN patients following antibody deposition [30,31,32] and their association with NFkB activation and macrophage polarization. Studies in strains of mice that spontaneously develop LN (MRL/lpr and B6/lpr) and in mice treated with pristane confirmed the implication of oxidants [33].

A main focus was the metabolic processes involved in ROS generation and factors involved in their regulation (such as Nrf2) that also influence the anti-inflammatory defenses in the kidney. A recent study [34] demonstrated that Nrf2 also modulates the formation of Neutrophil Extracellular Traps (NETs), which is one of the basic factors predisposing patients to the formation of autoantibodies in SLE and LN. In accord with the concept that Nrf2 plays an inhibitory effect on either ROS synthesis or autoantibody generation in LN, knockout mice have more inflammatory activity in glomeruli and display an increased susceptibility to developing LN [35]. Nrf2 implication in animal models of LN is also of interest in consideration of the fact that it may represent a potential target of new therapeutic options (see below).

Only a few studies investigated molecular factors involved in oxidative distress during LN in humans. However, in a significant study in a group of patients with LN, levels of Nrf2 in serum, in glomeruli, and mesangium were positively correlated with the estimated glomerular filtration rate (the protective effect in the kidney) and were also inversely correlated with SLEDAI, indicating diffuse protection for other inflammatory sites [32].

Overall, these observations suggest a complex mechanism in which antibodies represent the trigger of mechanisms targeting oxidative and inflammatory effects within glomeruli, ending with the production of oxidants [36,37]. Considering studies in mice and humans, it seems that even moderate up-regulation of Nrf2 activates both anti-inflammatory and anti-oxidant systems and might represent the target for new additional therapeutic strategies in SLE [38,39].

## 4. Anti-SOD Antibodies: Clinical Association and Significance

Circulating anti-SOD antibodies have been identified in patients with MN and LN in addition to the antibody causing the disease (Table 1) [40,41,42]. In both conditions, levels of anti-SOD antibodies have been associated with poor clinical outcomes [41,42]. Based on clinical observations in large cohorts of patients with MN, a scheme has been proposed according to which MN patients were sub-divided into groups according to the circulating levels of the primary antibody (anti-PLA2R1) and of anti-SOD antibodies; in the long term, the group with the highest levels of anti-SOD also had high levels of proteinuria, whereas a large part of those patients negative for any antibody had complete remission after two years (Figure 1). Patients who had both anti-PLA2R1 and anti-SOD2 antibodies in serum had the worst outcome. It was concluded on this basis that anti-SOD represents an independent predictor of poor clinical outcome [41]. In consideration of other studies (see above) indicating up-regulation of the expression of SOD by podocytes in vitro and in vivo after exposure to antibodies considered the trigger of MN (i.e., anti-PLA2r and anti-THSD7A) and in glomeruli of patients with MN, these data can be interpreted as a multi-step mechanism in MN in which the first antibody induces oxidative stress that is blocked by SOD and a second phase in which anti-SOD antibodies neutralize SOD, with detrimental effects on the kidney.

Circulating anti-SOD antibodies have also been found in LN, an observation that amplifies the spectrum of clinical conditions in which these antibodies potentially play a role [42]. In LN, at the onset of renal symptoms, the levels of anti-SOD were significantly higher compared to either controls or patients with SLE and then normalized after immuno-suppressive treatments. The kinetics of the reduction in anti-SOD2 was correlated with the reduction in proteinuria. Given the association of anti-SOD antibodies with proteinuria and in consideration of the studies indicating up-regulation of the expression of SOD by podocytes in vitro and in vivo, the same multi-step mechanism was proposed for LN which was proposed above for MN, in which the first antibody (anti-dsDNA, anti-ENO in the case of LN) induces oxidative stress followed by up-regulation of SOD and anti-SOD antibodies. A possible definition for this group of auto-antibodies occurring in a second phase of disease is ‘second wave’ antibodies, since this term, beyond the temporal characteristic, also strengthens the concept of an adjunctive effect that may overlap the initial and pathological function of the primary autoantibody that is directly involved in the pathogenesis of the disease. The conclusion is that in autoimmune glomerulonephritis, the basic mechanism induced by autoantibody deposition is the activation of oxidative stress and SOD is an essential element to counteract oxidants and maintain kidney homeostasis. Its blocking by anti-SOD antibodies is detrimental to tissue evolution to sclerosis and fibrosis. The poor outcome of patients with circulating anti-SOD antibodies is a direct demonstration of the importance that the anti-oxidative function of SOD plays in this circumstance.

## 5. Anti-GST Antibodies in Post-Transplant Glomerulopathy

In humans, cytosolic glutathione S-transferases (GSTs) are a family of enzymes with remarkable structural similarities and overlapping roles. Their primary function is to catalyze the conjugation of the reduced form of GSH with a wide variety of electrophilic molecules of endogenous and exogenous origin and facilitate, in this way, detoxification. Pesticides, herbicides, carcinogens, and variably derived epoxides are only a part of the substances that are cleared by GSH.

Besides the detoxifying function, GST enzymes play a key anti-oxidant function by facilitating S-glutatyionilation of proteins such as Peroxiredoxins (Prdxs) that use the thiol group of GSH to catalyze the reduction of H_2_O_2_, and other alkyl hydroperoxides [44]. Deletion of GSTM1 increases susceptibility to end-stage renal disease in mice [45].

The clinical importance of anti-GST antibodies is in recipients of a kidney transplant who may develop these antibodies as a part of a more generalized autoimmune response against the graft, also known as ‘Antibody-Mediated Reaction’ or AMR. In these patients, the presence of high levels of anti-GST theta1 antibodies is associated with an increased risk of graft loss [43]. As for the anti-SOD antibodies described above, a two-wave mechanism with a trigger effect in the kidney of oxidants followed by GST stimulation should play a role as a stimulus of anti-GST antibodies.

## 6. Potential Anti-Oxidant Approaches

Anti-oxidants represent a category of substances that have been widely investigated in experimental renal pathologies and, in some cases, in human beings. Most belong to the category of ‘natural compounds’ and often correspond to nutrients contained in diets; the diet may also have beneficial effects per se. Studies on mechanisms have indicated that many natural compounds modify factors implicated in the modulation of distress, such as nuclear factor erythroid-derived 2-like 2 (Nrf2), which is a transcriptional factor directly involved in the regulation of the antioxidative response [34,35]. Finally, a few drugs have been utilized for preventing, blocking and possibly reverting oxidative renal lesions. Even though the separation among these categories may appear artificial, it seems functional to clarity. The passage to humans should be easy given the extremely low toxicity of the compounds utilized in animal models.

One consideration that should be kept in mind in the choice of ‘therapeutic’ antioxidants is if the effect is linked with up-regulation of SOD, as documented in some cases [46,47]. In theory, if SOD is increased, the amount of anti-SOD antibodies may increase as well, leading to a null effect.

*Diet.* Monitoring nutritional status along with adequate diet control diminishes oxidative stress and inflammation and represents the basis for improving SLE prognosis and quality of life for patients. The mainstay for an optimal anti-oxidant diet in LN is a restriction of either calories and fat and their substitution by mono- and polyunsaturated fatty acids (mainly ω-3 and ω-6); minerals (selenium, zinc, and copper) and vitamins should be increased in parallel. All these prerequisites that characterize the traditional Mediterranean diet have been proven beneficial in SLE by diminishing oxidants and inflammation.

*Nutrients.* The difference between ‘nutrients’ and ‘natural substances’ described below is the frequency and the amount of a substance that is utilized in diets. Olive oil has a special relevance as a nutrient since it is a major component of the Mediterranean diet and is of easy availability in almost all Western countries. In BALB/c mice with pristane-induced lupus nephritis, olive oil was found to reduce immunocomplex deposits, glomerulosclerosis as well as interstitial inflammation and fibrosis, resulting in a reduction in proteinuria and amelioration of renal function [46].

A second important antioxidant in the Mediterranean diet is Resvetrol, which is a component of red grapes and red wine [47].

*Natural substances.* Natural substances, not necessarily contained in diets, may have an antioxidant activity.

Curcumin (an active ingredient of curry) is utilized in several eastern countries and in South America as an adjuvant for modifying either the color or smell of food and is the most popular compound of this group. The Western diet does not commonly utilize curcumin. In a randomized placebo-controlled study, curcumin significantly reduced proteinuria, hematuria, and systolic blood pressure in patients with relapsing or refractory LN without side effects and, in combination with vitamin D, it was effective in reducing disease activity, IL-6 levels, and fatigue severity [48].

A mix of natural compounds utilized in Chinese medicine to mitigate the renal effect of arterial hypertension is *Taohongesiwu,* obtained by decocting six herbs (*Prunus persica*, *Angelica sinensis*, *Curthamus tinctorius*, *Ligusticum Chuangxiong*, *Rehmannia glutinosa*) [49]. Beyond hypertension [49], the interest oi *Taohongesiwu* has been more recently extended to several renal pathologies including IgA nephropathy and renal fibrosis [50]. It has been suggested that the molecular mechanism of *Taohongesiwu* involves inhibition of ferroptosis and p53/Nrf2/p21 regulation [49].

Celastrol, a natural extract from the root of *Tripterygium wilfordjj,* is another compound utilized in Chinese medicine as an anti-oxidant with neuroprotective activity in brain ischemia [51]. Recent studies documented a direct protective effect of Celastrol on Nrf2 degradation in astrocytes [52].

*Antioxidant drugs*. Acetyl cysteine is the most important antioxidant utilized in medicine to protect the kidney from compounds with potential oxidant activity such as those utilized in several radiology approaches, especially iodate contrasts [53].

Melatonin (extracted from the pineal gland) is another medical substance with recognized anti-oxidant activity. It protects the kidney from glomerular lesions induced by pristane in BALB/c mice, thus suggesting a general positive effect in lupus nephritis [54].

Several other substances (i.e., edaravone, allopurinol, ascorbic acid) exhibiting neuroprotective effects have been utilized to improve neurological function in patients affected by ischemic stroke [55,56].

Experience with specific antioxidants in human pathologies is extremely limited, probably because the design of randomized clinical studies to support an adjunctive role of these substances is difficult in conditions that require a high number of drugs as basic therapy.

## 7. Conclusions and Perspectives

Autoimmune glomerulonephritis and the antibody-mediated reaction that follows solid organ transplantation represent two numerically important types of renal pathologies, with clinical impact, for the propensity to develop renal failure [57]. In both cases, the activation of oxidative stress and up-regulation of enzymes with anti-oxidant activity trigger a second phenomenon characterized by the formation of autoantibodies to anti-oxidant enzymes (anti-SOD and anti-GST, respectively). Their presence in circulation strongly correlates with poor organ outcome and their detection may serve to identify patients with poor prognoses.

As SOD and GST are cytosolic proteins, their recognition before antibody generation likely takes place in the extracellular compartment after cell apoptosis of circulating leukocytes or macrophages. Moreover, it is known that there is an interchange of GSH/GSSG and related enzymes from the intracellular compartment to serum, especially during the inflammatory process. We hypothesize that anti-SOD and anti-GST antibodies that are produced outside the cell can be highly toxic inside. We should learn more about the time-course changes of the oxidant/anti-oxidant balance and the formation of antibodies targeting the two enzymes above. Defining possible protective strategies is the final objective.

A simple approach could be to utilize anti-oxidative substances as soon as the diagnosis of either autoimmune glomerulonephritis or autoimmune-mediated transplant reaction is carried out. Testing circulating levels of anti-SOD/anti-GST antibodies would modulate the anti-oxidant approach. Generic anti-oxidants such as N-acetyl-cysteine may be utilized very rapidly in case–control studies based on the low number of adverse effects. Bioactive nutrients (including olive oil, resveterol, ascorbic acid, tocopherols, carotenoids, Zn, and Fe) and nonnutrients (such as phenolic acids, flavonoids, triterpenoids, and isothiocyanates) are further options. Melatonin (extract of pineal gland) is a compound utilized with positive effects in animal models of SLE and LN [58,59,60].

Owing to the very low number of side effects, the above approaches should become of easy and rapid application in human beings.

## Figures and Tables

**Figure 1 antioxidants-13-01519-f001:**
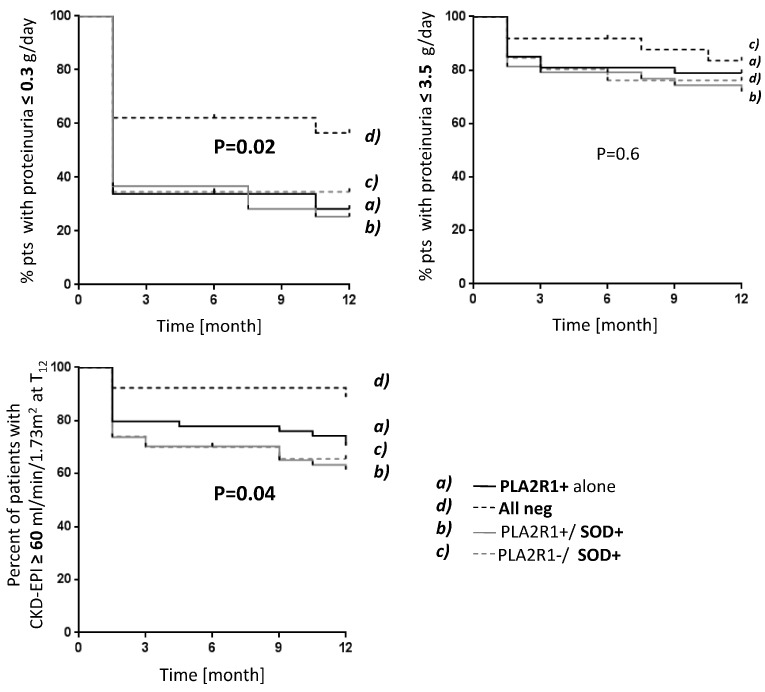
Patients with MN (285 overall) were studied at diagnosis and during a long follow-up of 46 months with multiple clinical approaches. At diagnosis, 182 (64%), 8 (3%), and 95 (33%) patients were anti-PLA2R1+, anti-THSD7A+ and double-negative, respectively. Kaplan–Meier analysis showed that anti-PLA2R1 and anti-SOD2 antibodies at diagnosis were each independently associated with poor clinical outcomes. Combined positivity for anti-PLA2R1 and anti-SOD conferred maximal risk.

**Table 1 antioxidants-13-01519-t001:** Summary of the literature on renal SOD up-regulation and anti-SOD2/anti-GST antibodies in patients with autoimmune Glomerulonephritis and post-transplant antibody-mediated reaction. Abbreviations: SOD, Superoxide Dismutase 2; GST, Glutathione-S-transferase; THSD7A, Thrombospondin-type1-domain-7A; AMR, Antibody-Mediated reaction; MN, Membranous Nephropathy; LN, Lupus Nephritis.

Area of Interest	Main Finding	Ref.
**SOD regulation**	- SOD is up-regulated in glomeruli of MN patients.	- Buelli et al., 2015 [25]
	- anti-THSD7A induces the synthesis of SOD2 by podocytes	- Thomas et al., 2026 [26]
**Anti-SOD autoantibodies**	- Anti-SOD2 in MN	- Murtas et al., 2012 [40] - Ghiggeri et al. 2020 [41]
	- Anti-SOD2 in LN	- Angeletti et al., 2021 [42]
**Anti-GST autoantibodies**	- Anti-GST are high in patients with AMR	- Comoli et al., 2022 [43]

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
