# Peer review of "Antibodies Against Anti-Oxidant Enzymes in Autoimmune Glomerulonephritis and in Antibody-Mediated Graft Rejection"

_antioxidants, 2024, doi:10.3390/antiox13121519_

Round 1

Reviewer 1 Report

Comments and Suggestions for Authors

Great perspective and great suggestion for immune mediated glomerulonephritis. I have only some minor suggestions.

1.     Authors suggest that patients had better receive the anti-oxidant agents at earlier stages, to avoid the production of anti-SOD or anti-GSH antibody, but they must be very careful to choose the anti-oxidant agents. If these agents can enhance the SOD or GSH expression, it may cause more high antibodies.

2. Nrf2 activator might be a good choice, and authors had better add this content and discuss its possible beneficial effect.  I suggest authors read the following publications and cite them in the revised manuscript.  

Celastrol targeting Nedd4 reduces Nrf2-mediated oxidative stress in astrocytes after ischemic stroke. J Pharm Anal. 2023 Feb;13(2):156-169. doi: 10.1016/j.jpha.2022.12.002.  

Inhibition of ferroptosis ameliorates hypertensive nephropathy through p53/Nrf2/p21 pathway by Taohongsiwu decoction: Based on network pharmacology and experimental validation..J Ethnopharmacol. 2023 Aug 10;312:116506. doi: 10.1016/j.jep.2023.116506 

Protective effects of catalpol on cardio-cerebrovascular diseases: A comprehensive review.  J Pharm Anal. 2023 Oct;13(10):1089-1101. doi: 10.1016/j.jpha.2023.06.010

Author Response

Please find the response in the attachment.

Reviewer 2 Report

Comments and Suggestions for Authors

I appreciate the paper by Bruschi et al. regarding the role of anti-oxidant antibodies in autoimmune glomerulonephritis. I only suggest minor comments to improve the quality of the manuscript and its comprehension.

- Some different parameters have been tested to assess better/adverse prognosis in idiopathic membranous nephropathy (e.g., podocyturia, 10.1038/s41598-020-73335-2). May the detection of anti-oxidant antibodies be combined with this assessment to identify early patients with negative outcomes, immunological activation, and more aggressive immunosuppressive management? Please consider and comment on this.

- Consider a brief table to summarize the reported studies with more meaningful results for each condition.

- Correct minor typos (e.g., that that on Page 2, Line 68; line interruption between 259-260)

Author Response

(The authors gave the same response as above.)

Reviewer 3 Report

Comments and Suggestions for Authors

Oxidative stress may play a role in autoimmune glomerulonephritis through the up-regulation of oxidant/antioxidant systems following autoantibody deposition, with anti-SOD antibodies potentially worsening clinical outcomes in conditions like membranous nephropathy and lupus nephritis. Similarly, renal transplant patients may develop anti-GST antibodies, leading to rejection, and thus antioxidant therapies proven effective in animal models might help modulate these responses in human kidney diseases.

Overall, the context is clear and smoothly described, and we believe it provides an introduction to a new perspective.

Major comments

1. The main claim regarding anti-SOD antibodies and anti-GST antibodies is whether there is reliable clinical data supporting their role as biomarkers or therapeutic targets. If such data exists, it should be cited; if not, it should be stated as absent, and future directions should be suggested accordingly.

2. At least one figure explaining perspectives should be developed.

Minor comments

1. The definition of “second wave of antibodies” should be explained.

2. All abbreviations should be spelled out in full when first listed. SOD, GST, GSH, NADH, FADH, NADPH, GSSG, etc.

3. Keywords: SOD should be superoxide dismutase. GSH should be glutathione. GST should be glutathione synthetase.

Author Response

(The authors gave the same response as above.)
